# Cardiomyocyte Proliferation from Fetal- to Adult- and from Normal- to Hypertrophy and Failing Hearts

**DOI:** 10.3390/biology11060880

**Published:** 2022-06-08

**Authors:** Sanford P. Bishop, Jianyi Zhang, Lei Ye

**Affiliations:** Department of Biomedical Engineering, University of Alabama at Birmingham, Birmingham, AL 35233, USA; jayzhang@uab.edu (J.Z.); lye@uab.edu (L.Y.)

**Keywords:** fetal, neonatal, hyperplasia, physiologic hypertrophy, pathologic hypertrophy, sarcomere, failure

## Abstract

**Simple Summary:**

Death from injury to the heart from a variety of causes remains a major cause of mortality worldwide. The cardiomyocyte, the major contracting cell of the heart, is responsible for pumping blood to the rest of the body. During fetal development, these immature cardiomyocytes are small and rapidly divide to complete development of the heart by birth when they develop structural and functional characteristics of mature cells which prevent further division. All further growth of the heart after birth is due to an increase in the size of cardiomyocytes, hypertrophy. Following the loss of functional cardiomyocytes due to coronary artery occlusion or other causes, the heart is unable to replace the lost cells. One of the significant research goals has been to induce adult cardiomyocytes to reactivate the cell cycle and repair cardiac injury. This review explores the developmental, structural, and functional changes of the growing cardiomyocyte, and particularly the sarcomere, responsible for force generation, from the early fetal period of reproductive cell growth through the neonatal period and on to adulthood, as well as during pathological response to different forms of myocardial diseases or injury. Multiple issues relative to cardiomyocyte cell-cycle regulation in normal or diseased conditions are discussed.

**Abstract:**

The cardiomyocyte undergoes dramatic changes in structure, metabolism, and function from the early fetal stage of hyperplastic cell growth, through birth and the conversion to hypertrophic cell growth, continuing to the adult stage and responding to various forms of stress on the myocardium, often leading to myocardial failure. The fetal cell with incompletely formed sarcomeres and other cellular and extracellular components is actively undergoing mitosis, organelle dispersion, and formation of daughter cells. In the first few days of neonatal life, the heart is able to repair fully from injury, but not after conversion to hypertrophic growth. Structural and metabolic changes occur following conversion to hypertrophic growth which forms a barrier to further cardiomyocyte division, though interstitial components continue dividing to keep pace with cardiac growth. Both intra- and extracellular structural changes occur in the stressed myocardium which together with hemodynamic alterations lead to metabolic and functional alterations of myocardial failure. This review probes some of the questions regarding conditions that regulate normal and pathologic growth of the heart.

## 1. Introduction

Growth of the heart from fetal to adult and pathologic hypertrophy after conversion from fetal cardiomyocyte (CM) hyperplastic growth to early neonatal hypertrophic growth has been studied by many investigators from the early studies of Karsner, Saphir and Todd [1], and Richter and Kellner [2]. Most investigators have concluded that all increase in cardiac mass after the early neonatal period is due to an increase in size, and hypertrophy, of CMs [3,4,5,6,7,8]. However, several recent reports have renewed interest in the possibility that CMs in adult animals may be stimulated to divide and aid in the repair of injury [8,9,10,11,12,13,14]. Recent reports in neonatal P1 mice [15] and pigs [16,17], demonstrated that complete repair of apical resection occurs without scar formation, but not at later times after myocytes have entered their hypertrophic growth phase. Developmental structural changes in the sarcomere, the functional unit of the CM, play a critical role in the inability of the cells to continue dividing and in the functional status during normal growth and overload conditions. The conversion from single nucleated hyperplastic CMs to binucleated hypertrophically growing CMs is a major adaptation in the growth cycle [18,19]. During fetal, neonatal, and adult growth structural, biochemical, and physiologic changes occur which control the inability of CMs to continue dividing and to function normally [11,14,20,21,22].

During the late fetal or early neonatal period, CMs undergo one final nuclear mitosis and cell division resulting in a near doubling of cell number of single nucleated cells, 1 × 2N [6,7,18,19,22,23]. Subsequent nuclear mitosis in some CM is not followed by cytokinesis, resulting in 50–95% binucleated CMs, 2 × 2N, in various subprimate animal species [6,18,19,22,23,24,25,26]. In humans [27] and subhuman primates [28], the doubling occurs more slowly, encompassing as much as the first two decades of growth. In both human and subhuman primates, DNA replication results in polyploidy of single nucleated CMs, 1 × 4 + N, double nucleated, 2 × 2N cells representing only about 25% of all CMs [27,29,30]. CMs increase in volume from the time of birth to adulthood by a factor of 15–20 times in parallel with the increase in cardiac mass [6,7,18,31,32,33]. With added stress due to pressure, volume overload, or ischemic myocyte loss, CM volume may increase to twice or more than the normal adult volume [7,34,35,36]. Whether adult CMs can reproduce themselves or be induced to divide to significantly aid in the repair of an injured myocardium continues as an unresolved issue.

From mice to the largest animals including humans, through both normal cardiac growth and pathologic hypertrophy, the primary contractile unit of the CM, the sarcomere, undergoes similar developmental and functional changes. From early fetal to neonatal and on to adult morphology, the structural and contractile proteins of the sarcomere undergo alterations ranging from dissociation in early fetal, to a highly structured and stable state after the neonatal period. The explanations for loss of myocardial function in pathologic hypertrophy and congestive heart failure include alterations in structure, hemodynamics, oxygenation, biochemical, and others. From the time prior to the advent of modern high-resolution electron microscopy, confocal microscopy, immunostaining, and physiological and biochemical methodology when the heart was considered to be both a functional and morphologic syncytium to the present time, major advances in the understanding of CM structure and function have been achieved. However, despite many both clinical and experimental animal and laboratory studies, the interaction of the many factors responsible for the altered function, structure, and reproductive capacity of CMs exposed to pathologic stresses leading to additional hypertrophy and failure remains unexplained.

## 2. Fetal and Early Neonatal Hyperplastic Cardiomyocyte Growth

The earliest beating CMs appear during mid-gestation in the rat and mouse when the cardiac loop stage is completed, about E10-12. During this mid-gestational rapid growth period, the heart has completed its growth from primordial cells, entered hyperplastic growth, and started beating [37,38]. CMs and endothelial cells are rapidly dividing as the heart increases in mass. However, both DNA synthesis and increase in heart weight are dramatically slowed during late fetal development prior to the conversion from hyperplastic growth to hypertrophic CM growth [22]. Both DNA synthesis and heart weight then rapidly increase during the early neonatal period as CMs exit hyperplastic growth and become binucleated or polyploid as in primates. DNA synthesis then declines to very low levels as conversion to hypertrophic growth is completed during the first weeks of neonatal growth [6,18,22,36,39,40]. Concurrently, several regulatory genes of DNA synthesis including CYCD1, CYCD3, CDK4, PCNA, Hippo-Yap1, and others are highly expressed during the late fetal and early neonatal period [20,22,37,41,42,43]. The conversion from fetal hyperplastic growth to neonatal hypertrophic cell growth is marked by a number of genes including Gax [22], Meis 1 [41,43], Hippo-Yap1 [43], and others reviewed by Paradis [42] and Brooks [44].

Ultrastructural studies of fetal myocardium in cell cultures [45,46], chicken embryos [47], rats [48,49], mice [50,51], and in an in vivo cell culture model of E12 rat myocardium growing in oculo [49] have identified incomplete developing structures in CMs and the endothelial and connective tissue of interstitial space. Endothelial cells forming new capillaries and interstitial connective tissue are sparse during mid-gestation, becoming more prominent near term [38,47,48,49,50,51]. Incomplete myofibrils and sarcomeres of E12 rats are loosely arranged around immature nuclei and mitochondria in spherical shaped CMs with abundant clear cytoplasmic space as shown in Figure 1.

As cardiac size increases throughout gestation with increasing workload, by birth, subsarcolemmal myofibrils have incomplete sarcomeres with Z-bands, actin, and myosin filaments but no M-bands [18,36,49,51]. Although not evident by electron microscopy, immunostaining techniques have demonstrated that the sarcomeric proteins are present in hyperplastic fetal CMs (4). As interventricular pressure gradually develops in late gestation to 20–70 mmHg in different species [31,52], the mid gestational rounded cells become elongated and spindle-shaped with myofilaments aligned under the sarcolemma (Figure 2). Figure 3 illustrates the sarcomeric Z-bands with attached actin filaments aligned with immature sarcolemmal dense filament adherens and developing intercalated discs (ICD). Tritiated leucine autoradiographic studies of neonatal growing myocardium localized leucine uptake strongest at the cell periphery at adherens discs and Z-bands [53,54].

There is a close association of nuclear rough endoplasmic reticulum, RNA ribosomes, and Golgi with newly formed undeveloped mitochondria at nuclear pole regions, suggesting the generational relationship between nucleus and mitochondria [55,56,57]. Once formed, mitochondria using their own mRNA are able to increase in size and form new cristae from the inner membrane which harbor the proteins necessary for calcium transit, the citric acid cycle, oxidative phosphorylation, and production of ATP as well as other functions [58,59]. Mitochondria also increase in size by fusion with other mitochondria and reproduce themselves by fission, splitting off smaller mitochondria [55,56,57,59].

Mitoses are frequent in fetal and early neonatal hearts [18,31,38,45,46,47,49] with well-formed centrioles and spindles [60,61,62,63] as illustrated in Figure 4. Loss of the centrioles after birth and the relocation of spindles to the nucleus are important factors in the inhibition of further CM division [61,62,63]. During the fetal and early neonatal hyperplastic phase of growth during nuclear mitosis, the sarcomeres are totally disrupted into the actin, myosin, and other sarcomeric proteins, and then reassembled in the daughter cells [46,48,60,64]. Immunostaining methods have documented the dissolution and restructuring sequence of these proteins during cell division [63,64]. Disassembly of the sarcomere during cytokinesis starts with the Z-band proteins sarcomeric α-actinin and titin, followed by cardiac α-actin and myosin, and finally by the myomesin M-band proteins [63,64]. A similar sequence of assembly is supported by electron microscopic evaluation of sarcomere genesis in growing or hypertrophying cardiac muscle [18,36,38,46,47,48,49]. The nuclear envelope and Golgi are also completely disrupted and then reassembled in the daughter cells [65]. Mitochondria during mitosis undergo fission, regulated by dynamin-related GTPase Drp1, resulting in many small mitochondria passed on to the daughter cells and then reassembled into functioning mitochondria [59,65] (Figure 4).

During the fetal to early neonatal period, connective tissue components are also present, though poorly visualized by EM. Terracio [66] has shown that the major components of fibronectin, collagen types I, III, and IV, and laminin are present in the fetal myocardium and persist through neonatal active cell growth and decline during stable adult life. New capillaries are forming with a capillary to fiber ratio of about 1:4 in neonatal myocardium, comparable to the adult capillary to fiber diffusion distance [18,67,68,69]. As CMs increase in size with growth, capillaries rapidly proliferate in the early neonatal period, proliferation slowing as cells reach adult maturity [70]. Capillary to myocyte diffusion distance and thus oxygen availability is maintained throughout growth from neonatal to adult to meet changing metabolic requirements [71,72].

Energy production during the early and mid-stage of fetal development is mainly from glycolysis, although oxidative metabolism provides some of the ATP for energy [73,74]). In the E12 rat, newly formed mitochondria are small and centrally located around the nucleus and have less dense cristae than in neonatal or older myocardium [38,45,47,49,54]. The transition of energy production from mainly glycolysis to a gradual increase in oxidative metabolism and the maturational development of mitochondria as the fetus nears term have been reviewed by Ascuitto [58]. As fetal gestation progresses toward birth, mitochondria become more numerous with more fully developed cristae [49,54]). In P1-3 rats and dogs, numerous small mitochondria are clustered mainly at the nuclear poles with a few in subsarcolemmal locations. Some mitochondria have increased in size and cristae are well developed, reflecting the conversion of major energy production to oxidative metabolism [18,36,49,54].

At the transition from fetal to neonatal life of altricial animals, rats, mice, dogs, and cats, virtually all CMs are mononucleated [6,7,18,19,31,36], while over 50% of multinucleated cells are present in sheep [25,26] and 11% in pigs [24], and possibly other precocial animals as well as in humans [33]. Sarcomeres are structurally incomplete, lacking EM visible M-bands. Mitochondria are centrally located, a single layer of myofibrils lines the periphery, and the CMs still lack sarcoplasmic reticulum and T-tubules. During the late fetal and early neonatal period while intracellular cytoplasmic structures including sarcomeres and mitochondria are maturing but still incomplete, dissociation is still possible allowing mitosis followed by cytokinesis. Although there are many factors involved in the loss of CM’s ability to continue dividing, the maturation of both intracellular and extracellular components would appear to be a major factor in the hypertrophically growing cells’ inability to divide.

## 3. Conversion from Fetal to Neonatal Cardiomyocyte Growth

### 3.1. Morphology

Birth is accompanied by dramatic changes for the fetus as it emerges from the controlled temperature, nutritional and oxygenation environment provided by placental support from the mother to existing on total external support. Both body weight and heart weight are increasing only slowly during the final fetal days as fetal cell number declines just before birth [22,25,26]. Body and heart weight both rapidly increase following birth, doubling within the first few days to weeks of post-natal life [6,18,22,25,31]. Due to the placental blood circulation, and patent ductus arteriosus, left and right ventricles have nearly equal mass at birth [25] but with the switch to pulmonary circulation and closure of the ductus arteriosus, the right ventricle has very little growth early after birth while the left ventricle (LV) rapidly gains mass. By P6 in rats [31], P10-12 in dogs [18], P14 in pigs [24], and P20 in sheep [25], right ventricular (RV) growth keeps pace with LV growth.

Pigs and sheep, and presumably other precocial hoofed animals have started the binucleation during the late fetal period [24,25,26]. Sheep start the conversion to binucleated CM during the last third of gestation, have a sharp drop in CM number just prior to birth, and have over 80% binucleated CMs soon after birth. Very little binucleation occurs after birth and the cell number of CM is stable soon after birth. The volume of CMs starts to increase at birth along with LV mass [25]. Pigs also start the conversion late in gestation. However, as opposed to sheep, at birth, only 11% of CMs are binucleated [24] and binucleation continues for several weeks [75]. Mice, rats, dogs, and other altricial animals have 100% 1 × 2N cells at birth, and conversion to binucleated CMs is delayed until P4 in rodents [19,31] and P10 in dogs [18,76].

### 3.2. Metabolism

At birth, blood oxygenation is suddenly increased from the relative placental supplied hypoxia of 25–35 mmHg during fetal life to the pulmonary circulation with 75–90 mmHg, metabolism converting from relatively glycolytic to free fatty acid oxidative to supply the ATP required for the rapidly increasing cardiac workload [58,73,74]. The transition from glycolytic to oxidative metabolism is not sudden, however, but appears to occur gradually during late fetal and early neonatal growth, and varies with species [41,58,74,77]. Sheep and pigs, for example, reduce glycolytic metabolism in the late fetal period as oxidative metabolism moderately increases and converts to full fatty acid metabolism soon after birth [58].

That these animals rely heavily on glucose metabolism until conversion to hypertrophic growth [41,58,74] has led to the hypothesis that oxidative metabolism may be linked to the loss of cardiac regeneration [41,77]. With the exposure to an oxygen environment after birth, mitochondria mature, producing needed ATP and as a side product, reactive oxygen species (ROS). In neonatal mice, ROS were significantly increased while CM was converting from hyperplastic growth to hypertrophic growth [77]. ROS activate DNA damage resulting in cell cycle arrest suggesting that ROS are actively involved in the conversion to hypertrophic cell growth. Supporting this theory, adult mice exposed to 7% oxygen for two weeks had decreased ROS, an increased CM proliferation, and increased myocardial function after myocardial injury [78]. The variable time frames for conversion to hypertrophic growth and timing of metabolic conversion to oxidative metabolism among species with varying gestation time, litter size, maturity at birth, time to weaning, age of sexual maturity, and other factors suggest a causal relationship, but explanations are not available [75]. The role of this conversion from glycolytic metabolism to fatty acid oxidative metabolism in the conversion to hypertrophic growing and non-replicating CMs would appear to be related but remains unclear.

### 3.3. Sarcomere Structure

A number of molecular changes occur to sarcomeres during the conversion from fetal hyperplastic to neonatal hypertrophic growth, variable in timing in different species. Critical to providing sarcomere stability are changes in the myofilament proteins, myosin and actin, the regulatory proteins, tropomyosin and troponin, and the cytoskeletal proteins, titin, myosin-binding protein, α-actinin, myomesin and M-protein [79]. Each of these sarcomere structural proteins has a fetal and adult form switching from the fetal form to the adult form around the time of birth and conversion from hyperplastic to hypertrophic CM growth [79,80]. The fetal forms control both contractile function and myocyte cell division and are downregulated as birth nears and the adult forms control function and growth in the hypertrophic CM [79,80].

The sarcomeric myosin heavy chain (MYH), β-myosin, prominent in the fetus, is replaced by the faster-contracting α-myosin as hyperplastic growth ends [79,80]. The α-cardiac and α-skeletal isoforms of actin also undergo isoform switching during conversion from fetal to neonatal growth [79].

Titin (connectin) is a protein in the sarcomere that connects the Z-band with the M-line in the center of the sarcomere. Titin plays a role in the passive tension of the sarcomere and is a template for myofibril assembly [80,81,82,83]. Five different isoforms of titin have been identified and there is a switching from fetal to adult forms concurrent with the isoform switching of myosin heavy chains [83].

During mid to late fetal growth, the slow skeletal muscle cardiac troponin I isoform ssTnI (TNNI1) is predominant [84,85]. During late gestation in sheep or by P3 in rodents, TNNI1 is downregulated and replaced with the mature sarcomeric protein isoform cardiac (c)TnI (TNNI3), completing a more stable form of sarcomeric structure [84,85].

Structural alterations in the sarcomere as well as increasing interstitial collagen connective tissue [86,87,88] would appear to be major factors in the loss of the ability of CMs to replicate. The increasing extracellular matrix in the postnatal heart has been shown to be important in preventing continuing CM cytokinesis [89,90]. Cardiac fibroblasts continue to proliferate during late fetal development and peak their proliferation around P4 in rats coincident with the initiation of binucleation [22]. The extracellular matrix including fibroblasts and collagen has both fetal and adult forms which play a role in the fetal CM division and after switching to the adult form in the neonatal period, promotes binucleation and cessation of cytokinesis [89,90]. The extracellular proteins SLIT2 and NPNT (nephronectin) have been shown to promote CM cytokinesis. Downregulation of these and other molecular changes in sarcomere structure in the late fetal and early neonatal period contribute to the increased ventricular passive tension and overall cardiac compliance resulting in a more stable form of the sarcomere [85,91].

Late fetal and early neonatal CMs have accumulated more mitochondria and fibrillar sarcomeric structure though still loosely organized to accommodate continuing mitosis and cell division. As body mass and blood pressure rapidly increase after birth, the growing heart adapts by increasing the structural organization of both CMs and interstitial components of connective tissue and capillaries.

### 3.4. Experimental Studies

In an attempt to unravel the myriad of possibilities that control the conversion from fetal hyperplastic growth to post-natal hypertrophically growing CMs that no longer are able to divide, many experimental studies have been published. Experimental interventions include genetically altered animals [92,93,94,95], fetal or early neonatal hypoxia [96,97,98,99,100], anemia [101,102], manipulations of metabolic or growth regulators [103,104,105,106], pressure, and volume overloads [107,108], and others. In general, experimental interventions during fetal or early neonatal growth tend to accelerate hyperplasia and alter the onset of hypertrophic growth. For recent reviews see [75,109,110,111]. Many of these interventions applied in adult animals exert their influence by the promotion of CM hypertrophy.

An example of the dual role played by experimental interventions is the protooncogene c-myc which mediates both proliferative and hypertrophic cellular growth in many cell types [112,113]. In a transgenic mouse with overexpression of the c-myc oncogene, the transgenic mice underwent one additional CM division during the fetal period. The transgenic mice had a 50–100% increase in P1 heart weight because of twice the number of CMs with little change in body weight compared to wild-type littermates [92,93,94]. In addition to the doubling of CM number during the fetal period, conversion to hypertrophic growth and binucleation began at P3 in the transgenic mice vs. P7 in the wild-type littermates, suggesting that the number of myocyte divisions may have some intrinsic control. In an adult tamoxifen-inducible myc transgenic mouse, myc activation-induced nuclear mitosis and a 41% increase in cardiac mass due to hypertrophy of the CMs [112], illustrating the differing effects of a stimulator on replicating vs. non-replicating CMs. Interestingly, in the c-myc transgenic mice which continued to express c-myc in adult mice, heart mass was greater than in wild-type due to increased cell number, but not due to increased cell size [92,93,94].

While much has been learned relative to the various physical, genetic, biochemical, growth factors, and other participants in the conversion from fetal to neonatal CM growth, there still remains the question of what determines the number of cell cycles required to reach the neonatal heart weight. Obviously, the number of cell cycles required to obtain the cell number needed for a neonatal mouse heart is much different from that of larger species such as a dog, horse, or human infant, even within the same species, for example, dogs. What controls the number of CM for a 10 lb. Chihuahua, a 40 lb. Golden Retriever or a 150 lb. Irish Wolfhound? The concept that fetal CMs have some intrinsic timer mechanism that determines when the appropriate number of cells has been achieved was earlier proposed using neonatal dog hearts [18] or neonatal rat hearts [19]. In an in vivo culture system, E12 fetal rat hearts implanted on the iris of an adult rat beat and grew as a solid mass and became vascularized and innervated [49]. CMs, initially loosely organized as early embryonic cells, continued to divide by karyokinesis and cytokinesis, adding functioning sarcomeres and mitochondria to new CMs. By 12 days in oculo, equivalent to a P3 neonatal rat, the CMs became binucleated with developed but disorganized myofibrils and incomplete sarcomeres. By five weeks in oculo, sarcomeres were structurally complete though myofibrils were disorganized. The in oculo hearts continued to beat through eight weeks providing a minor workload but in the absence of a pressure workload myocyte size increased only slightly from weeks 2–4 in oculo, but there was no further increase in size during the remaining time, suggesting that increasing arterial and ventricular pressure is required for CM to grow and become organized. Burton [114], using cultured embryonic rat CMs, reported that cells from different embryonic ages continued dividing for a determined number of cycles and then stopped dividing, possibly controlled by an intrinsic timer, genetic influences or perhaps influenced by some other conditions. The timing of conversion from hyperplastic fetal growth to hypertrophic CM growth starting in the late fetal period in sheep and pigs, and several days after birth in many other species, clearly indicates that other factors than birth are controlling the conversion [25,115]. These studies suggest that an internal genetic clock is controlling exactly how many cell cycles producing new CMs are required to produce appropriate neonatal heart size for each different individual.

## 4. Cardiomyocyte Neonatal Growth

In addition to the variable degree of binucleation occurring as cells transition from hyperplastic to hypertrophic growth, some species develop larger numbers of CM nuclei. Sheep have over 50% bi- or tetra-nucleated cells at birth and 80% as adults [26]. In pigs and giraffes, after conversion to hypertrophic growth, many CMs contain 4, 8, 16 and even more nuclei, 4+ × 2N [23,75,76,115,116,117]. In most mammals including primates, a small percentage of CMs, usually less than 1–2%, have four nuclei, sheep having up to 8%, due to a second mitosis of a double nucleated cell [18,24,25,26]. In humans, only about 25 percent are 2 × 2N, the remainder being polyploid single nucleated, 1 × 4 + N [27,29].

In all species, the process of bi-, multi-nucleation, or polyploidy results in the cessation of CM division, establishing the CM number for the life of the animal or person. The question is why in the heart when so many other tissues in the body continue to replicate throughout life? What is the function of having 2 or more nuclei or multiple sets of DNA in one nucleus for the maintenance of CM function? What is the relationship of the amount of DNA to cytoplasmic volume? Answers to these and other questions remain unexplained from observational descriptions of cell growth and continuing discovery of more factors regulating cell growth. One conclusion can be that since fetal dividing cells are immature, easily dissociated and contractile force is very weak, having a stabilized cell with mature sarcomeres and interstitial connective tissue, powerful contractile activity is an obvious advantage to satisfy the work demanded of the heart by the growing, and adult, animal. The addition of one or more sets of DNA clearly prevents further cell division and allows the CM to mature into a cell that can meet the increasing workload of growth. Having a set of CMs dissociating and replicating would appear to be a clear disadvantage to the work requirement of the heart [118]. An additional disadvantage of having uncontrolled CM division is that the heart might grow to an unmanageable size or even become cancerous [119,120].

What is the relationship of the number of nuclei to cell volume and function? In neonatal CMs with 1, 2, or 4 nuclei, cell volume is directly related to the number of nuclei [6,18,25,31,115,121]. Binucleated CMs continue to be approximately twice the volume of mononucleated cells even into adult life and cardiac hypertrophy [35,122]. In most animals studied, except the pig, CMs start to increase in volume as soon as binucleation occurs and the length to width ratio normalizes to the adult ratio within several days [18,19,122,123]. The pig differs in that binucleation continues through P15 followed by additional nuclei with CMs containing 4, 8, 16, and even more nuclei. Cell volume increases slowly through P30-60, mainly by the increase in cell length and only a minor increase in cell cross-sectional area. After two months of age, CMs increase volume by both cross-sectional area and length [115]. In all forms of nucleation, mono-, bi-, or multi-nucleated, CM volume increases from the neonatal mononucleated stage of 1–2000 µm^3^ to the adult binucleated size of 25–40,000 µm^3^ with the same number of nuclei.

An obvious conclusion from observation of the neonatal CM with a direct relationship of cytoplasmic volume to nuclear number is that increased cytoplasmic volume requires more DNA. However, the fact that CMs increase volume many times over with the same nuclear number questions that relationship. The process of bi-, multi-nucleation, and polyploidy is inhibitory to cell division in CMs [124,125], but what is the role of having multiple copies of DNA in one cell? Recent studies have provided evidence that the multiple copies of DNA may not all be performing the same function. One nucleus may be involved in protein synthesis and some other function assigned to a second nucleus, or possibly just dormant [126]. That multiple nuclei are not all participating in cell function similarly, is also supported by the finding that the transcriptomes of mono and binucleated CM of neonates are not different [120]. However, in aortic constricted rats, the hypertrophied CM did have increased transcriptomes relative to non-constricted controls [120]. In overloaded human hearts, polyploidy of single nuclei also supports the concept of the stressed CM requiring additional protein synthetic activity [121]. Despite these recent advances in understanding the role of multinucleation or polyploidy in CM, there remain many unanswered questions about nuclear control of the cell.

## 5. Post-Natal to Adult Growth

In the early post-natal period as CM become binucleated, CM volume increases proportionately with an increase in heart weight [31,32,127]. Within 1–4 weeks after birth, CMs gradually acquire adult cell morphology and functional properties. Cell volume increases by the addition of new sarcomeres formed from filament attachment areas of developing intercalated discs, which move from lateral regions to the cell ends, apparently the result of increasing contractile force exerting force on the ICDs (Figure 3). As new myofibrils are added laterally, existing myofibrils are moved inward, increasing the width of the CM. Sarcomeres are also added to the ends of existing myofibrils from intercalated disc filament attachment areas to increase CM length. Sarcomeres of adjacent myofibrils remain aligned and connected at the Z-bands. Sarcomeres become fully formed with structural proteins including titin and myomesin [18,85,91]. Mitochondria are added in the nuclear area, proliferate, increase in size and move to the sub-sarcolemmal area and along myofibrils [65]. By P11 in rodents as well as in other species, as CM start to enlarge, T-tubules and sarcoplasmic reticulum appear allowing calcium access to the inner cytoplasm and sarcomeres [18,128,129,130]. With the completion of the maturation of sarcomeres, mitochondria, intercalated discs, sarcoplasmic reticulum, and T-tubules, the growing CM assume the electromechanical functions of adult cells [130,131,132,133,134] (Figure 5).

Paralleling the structural and functional development of CM, the interstitial components of capillaries and extracellular matrix are also increasing. As CM increase in size, endothelial cells and smooth muscle cells proliferate forming new capillaries and small arterioles. Post-natal binucleated CM are 4–10 µm in diameter and the capillary: fiber ratio is 1:4 [18,67,68,69,72,135]. As CM increase in diameter and length, capillaries increase in number and length reaching the 1:1 ratio of capillaries: fiber in the adult [67,68,69,70]. Thus, diffusion distance from capillary to CM is constant throughout growth, maintaining constant oxygen availability [71,72].

Fibrocytes are sparse in the early neonatal period but rapidly increase as CM start to grow and produce a network of collagen fibers forming a solid scaffold around CM, capillaries, and larger vessels [86,87,88]. The fibrocytes and developing extracellular matrix play important roles in inhibiting cytokinesis and stabilizing cardiac structure, and consequentially preventing CM replication [89,90].

In mammalian species, as normal neonatal developmental growth is completed and CM enlarge, an increase in body weight and heart size stabilize at the adult level and remain relatively constant throughout the normal life span [18,136]. Factors that determine final body, heart, and CM size are unclear but may include gestational time, age at weaning, age at sexual maturity, nutrition, genetic factors, and other physiological functions [75,137].

Throughout the growth to adult size and adult stabilization, new proteins are continually added to increase cell size and to replace proteins that have definite half-lives [121,138,139]. A low rate of new CMs, presumably from preexisting 1 × 2N cells has been found in both humans and mice [29,33,140]. However, the rate is very low, about 1% per year, clearly not enough to provide significant CM replacement. Hypertrophy of remaining CMs does occur after the loss of a portion of the myocardium due to ischemic damage or other major cell loss [141,142]. Additional pressure overload, volume overload, hemodynamic and biochemical alterations, also result in major structural and functional alterations of the myocardium.

## 6. Effects of Additional Stress on Cardiomyocytes

With additional hemodynamic loads on the growing or adult heart, the CMs respond in an attempt to normalize wall stress and tension. According to the Law of LaPlace, wall tension is directly related to chamber pressure and diameter and inversely related to wall thickness. During increased stress on the myocardium, the CMs increase in volume to adjust to the increased wall tension with two forms broadly classified as physiologic or pathologic hypertrophy. With physiological or pathological conditions of either pressure or volume overload on the myocardium, wall stress increases, and the individual CMs respond by changes in cell size and shape but with differences in structure, function, and outcome.

### 6.1. Physiologic Hypertrophy

Physiologic hypertrophy is due to physiologic conditions such as normal growth, pregnancy, exercise-trained athletes, racing greyhounds, thoroughbred racing horses, wild hares, and migrating Alaskan Caribou. Due to the slower heart rate from the intermittent exercise, the stroke volume is increased to maintain cardiac output, and thus, a volume overload on the heart, resulting in a dilated LV chamber and normal or modestly increased LV wall thickness, eccentric hypertrophy. However, in contrast to the volume overload due to valvular insufficiency or arterial-venous (AV) shunts, the eccentric hypertrophy because of increased exercise, does not lead to cardiac failure. Physiologic hypertrophy due to isometric exercise such as weightlifting or throwing sports, results in intermittent pressure overload, leading to normal or even smaller LV chamber and increased wall thickness, concentric hypertrophy. Both forms of physiologic hypertrophy have normal or enhanced cardiac structure and function and are reversed when the stimulus is removed [143,144,145,146].

The gain in heart weight due to strenuous exercise in athletes is generally from 10–40%, but as noted by Linzbach [34], some exercising individuals may nearly double the normal heart weight, approaching the “critical mass” of 500 g. In a meta-analysis of echocardiographic studies of 1451 highly trained athletes in endurance training (running), strength training (weightlifting), and combined endurance and strength training (rowing, canoeing), LV end-diastolic diameters exceeded control values in all three groups. Septal wall thickness (and calculated LV mass) was increased from the control value of 8.8 mm to 10.5 mm (43%) for endurance-trained athletes, to 11.3 mm (65%) in combined training and 11.8 mm (53%) for strength-trained athletes [147]. In an additional MRI study of cyclists and controls, there were no significant abnormalities of cardiac function or metabolism in the trained athletes [148]. In another echocardiographic study of 947 trained athletes, 38% of hearts had LV end-diastolic diameters larger than the upper normal range, and mean septal wall thickness increased to 9.7 mm, only slightly above normal of 8.8 mm, but only 1.7% had LV wall thickness greater than 13 mm, the range compatible with hypertrophic cardiomyopathy [149]. Therefore, a small number of trained athletes may approach the “critical mass” of Linzbach, but the vast majority have less than 50% hypertrophy and retain normal structure and function.

It has been recognized for many years that some highly athletic animals have increased heart weights compared to others of the same species. Greyhounds and thoroughbred horses have been genetically selected to have body conformation ideal for running, and thus have a different body mass relative to size compared to other less athletic animals which account for some of the increase in relative heart weight to body weight (HW/BW). Thoroughbred horses have a HW/BW ratio of about 10–13% heavier than other horses [150] while greyhound dogs, due to their extremely lean body mass, have about 50% or more HW/BW than other breeds [150,151,152]. Race training adds an additional 5–10% HW/BW to both animals [150].

Experimental animal studies to evaluate physiologic hypertrophy have used swimming or treadmill exercise in mice [153,154], rats [155,156,157,158,159], pigs [160,161], and dogs [162,163,164]. In these animal studies, a moderate HW/BW increase of 5–25% was usually produced with over 30% increase in HW/BW with strenuous exercise [160]. In both the human athletes and animal studies, hypertrophy of CMs resulted in either no change or only minor alterations of myocardial structure including interstitial connective tissue and capillaries, function, and molecular or biochemical alterations [155,160,161,163,164].

### 6.2. Pathologic Hypertrophy

In contrast to the normal function and structure of the heart in physiologic hypertrophy due to exercise, increased stress and wall tension on the heart, due to pressure or volume overload, results in significant alterations often ending with cardiac failure. Pressure overload caused by hypertension, aortic or pulmonary outflow obstruction results in increased wall thickness and normal or smaller ventricular lumen, concentric hypertrophy. Volume overload, due to valvular insufficiency or AV shunts, results in a dilated LV and normal or mildly thickened LV wall, termed eccentric hypertrophy. Cardiac hypertrophy with or without congestive failure occurs in a wide variety of congenital or inherited anomalies in both humans [165]. and animals [166], including the widely used Spontaneously Hypertensive Rat (SHR) [123,167]. In addition to hypertrophy caused by pressure or volume overload, compensatory hypertrophy of the remaining myocardium results, following the loss of significant amounts of myocardium due to ischemia or diffuse myocardial disease. Smaller amounts of endo- or mid-myocardial myocardial loss result in concentric hypertrophy adjusting wall stress to normal while large transmural infarcts common in both humans and coronary ligated rats and mice, or diffuse cell loss, result in chamber dilation, eccentric hypertrophy, and eventual myocardial failure [141,168,169,170]. Inherited forms of hypertrophic or dilated cardiomyopathy in the absence of hypertension occur in humans [171,172], dogs [173,174,175], cats [176,177,178,179], and genetically altered transgenic mice [180,181,182,183,184,185,186]. In a comparative review of 28 adult patients, 10 dogs, and 51 cats with hypertrophic cardiomyopathy, the humans had more hypertrophy, myofibrillar disarray, and fibrosis than the dogs or cats [179]. Mean heart weight in humans was 548 g/kg, more than double normal with some patients’ heart weight over 1000 g. Dog and cat mean HW/BW was 45% and 33% greater than normal animals, respectively. In humans, dogs, and cats, hypertrophic cardiomyopathy leads to congestive heart failure and often, sudden death. There are differences in the initiation factors as well as the consequences of the different forms of pathologic hypertrophy.

From the early studies of Linzbach [34], and Meerson [187], as well as more recent studies [188,189,190,191], the concept has developed of progression through an initial compensatory stage with relatively normal or increased function to a second transitional stage of increasing functional and morphologic changes, to a final stage of irreversible decompensation and cardiac failure. The responses vary between pressure and volume overload.

## 7. Structural Response of Cardiomyocytes to Overload

The response of the individual CMs to volume or pressure overload is quite different. Clinical studies suggested [192,193,194], and several experimental animal studies [195,196,197,198,199,200,201] have confirmed earlier speculation, that volume overload with increased chamber diameter would result in an increase in cell length with a minimal increase or even a decrease in cell cross-sectional area, while pressure overload would result in an increase in cross-sectional area of myocytes with no change in length. Hemodynamic factors of ventricular chamber volume and pressure during systole and diastole determine CM morphologic changes attempting to normalize wall tension. Sarcomere length remains constant requiring a rearrangement of sarcomere units within the cell.

### 7.1. Volume Overload

In volume overloaded hearts, with a dilated ventricular chamber and increased end-diastolic pressure, myocytes not only become longer, but cell diameter is often only equal to, or even less, thanin normal hearts, leading some investigators to propose a “myofibrillar slippage” [202,203,204]. Rearranging the internal alignment of myofibrils would require some disruption of inter-myofibrillar Z to Z band desmin connections which normally keep sarcomeres in alignment. The loss of the lateral alignment of sarcomeres in dilated forms of hypertrophy occurs in human dilated cardiomyopathy [205,206] and in animal models of mitral valve insufficiency in dogs [207,208], tachycardia-induced ventricular dilation [209,210,211,212,213,214,215], and AV shunts [196,203] (Figure 6). This loss of sarcomere lateral alignment apparently contributes to the CM elongation with normal or reduced cross-sectional area in tachycardia-induced LV chamber volume with no increase in LV mass [212,213], or increased cross-sectional area in volume with an increase in LV mass [195,207,208].

Other contributing factors to the CM elongation in volume overload are alterations in the interstitial collagen structure. Several studies of experimental volume overload have found decreased attachment of laminin, fibronectin, and collagen IV to myocytes [212], and reduced extractable collagen from chronically dilated ventricles [213,216,217,218,219]. Scanning and transmission electron micrographs of volume overloaded myocardium reveal not only a loss of lateral alignment of sarcomeres but also a reduction in the normal interstitial collagen weave [209,212,215] (Figure 7). The loss of interstitial collagen fibers in chronically dilated myocardium would aid in the conformational alteration of the myocyte [203,212,213,215]. The role that various factors play in the control of the connective tissue including pressure and volume, the renin-angiotensin system, aldosterone, chymase, mast cells, metalloproteinases, procollagenase, and other factors has been discussed by Dell’Italia and Ruzicka [195,218].

### 7.2. Pressure Overload

During the early compensatory stage of developing pathologic hypertrophy due to pressure overload, the LV wall thickens to adjust to the increased wall stress. As the LV wall thickens, CMs increase in size by adding myofibrils in parallel, increasing CM width. In canine and swine models of progressive aortic stenosis due to banding of the ascending aorta at 2–8 weeks of age and studied 6–12 months later, there is a 70–90% increase in LV mass with normal end-diastolic pressure and minimal increase in connective tissue. Both humans and experimental animals with stable left ventricular hypertrophy (LVH) have normal to moderately increased basal coronary blood flow, mildly increased endocardial vs. epicardial blood flow at rest, and normal adenosine-induced coronary reserve [220,221,222,223,224,225,226]. During the stable stage of hypertrophy, the contractile function is retained and myocardial stiffness is normal [227,228,229]. Increased exercise-induced stress on stable LVH causes reduced subendocardial blood flow [225,230,231], and multifocal subendocardial fibrosis [232]). In this progressive aortic stenosis large animal model, some animals develop congestive myocardial failure with increased end-diastolic pressure and development of myocardial fibrosis [230,231,233]. Development of myocardial failure with increased end-diastolic pressure has increased resting myocardial blood flow and reduced subendocardial coronary flow reserve [230,231,233,234,235,236]. Although the total number of capillaries per unit area is somewhat reduced, increased capillary diameter and surface area plus CM expansion around capillaries provide adequate oxygenation [237,238,239]. Vascular morphology and capacity are normal in more recent studies supporting adequate vascular capacity for the hypertrophied LV, although earlier studies had variable results, reviewed by Tomanek [238]. However, in patients and animals with aortic valve stenosis or IHSS, and experimental animals with sub-coronary aortic stenosis where coronary perfusion pressure is normal while LV pressure is elevated, in contrast to supravalvular stenosis or systemic hypertension, there is subendocardial fibrosis and myofibril loss associated with reduction of normal coronary subendocardial blood flow [240,241,242].

## 8. The Role of Connective Tissue, Hemodynamic, Metabolic, Genetic, and Structural Alterations in Development of Myocardial Failure

### 8.1. Connective Tissue

The role of interstitial connective tissue in the development of stable or failing cardiac hypertrophy has received much discussion. Conflicting results are often the result of using different animal species and models. Many earlier studies of pressure overload-induced hypertrophy placed a constricting band on the pulmonary artery, thoracic or abdominal aorta of adult animals, producing a sudden increase in pressure resulting in focal areas of myocardial fibrosis [243,244,245]. The myocardial fibrosis was shown to be an artifact of the sudden onset of pressure, apparently producing a short-term oxygen supply/demand imbalance, causing multifocal myocardial ischemic necrosis immediately after the imposed afterload and not related to the development of hypertrophy [244,245]. An increase in interstitial connective tissue follows the early hypertrophy by several weeks to months in large animal models [190,246]. Similarly, in rats, isoproterenol-induced focal myocardial fibrosis is due to necrosis resulting from the sudden sharp drop in coronary perfusion pressure during the first hours of administration [247,248].

As described above, progressive gradual increase in systolic LV pressure from non-constricting bands placed on the ascending aorta of young puppies, kittens, or pigs results in a 50–90% or more increase in LV weight by 6–12 months of age, with minimal or no increase in fibrosis or increase in elastic stiffness until failure due to increased end-diastolic pressure occurs [217,221,223,224,228,246]. The use of different animal species and models has contributed to conflicting results relative to the role of fibrosis and myocardial stiffness. Many other studies have employed the progressive hypertrophy model by banding the ascending aortic of weanling rats, generally resulting in increased interstitial fibrosis and myocardial stiffness progressively increasing with time [204,245,249,250,251,252,253]. Isoproterenol administration to rats has been a widely used animal model to evaluate the effects of β-adrenergic stimulation on cardiac hypertrophy and the effects of fibrosis [247,248,254,255]. In the rat, isoproterenol results in multifocal areas of myocardial fibrosis as well as interstitial fibrosis complicating the interpretation of the effect of hypertrophy alone on myocardial function. Interestingly, chronic isoproterenol administration to mice did not produce focal necrosis and fibrosis and only minimal interstitial fibrosis [256,257,258], highlighting the variation in results from different species. The variety of various animal models studied to evaluate fibrosis have been reviewed by Ding et al. [259], and the molecular basis of fibrosis by Rockey [260].

### 8.2. Interstitial Collagen and Myocardial Stiffness

One of the hallmarks of pathologic cardiac hypertrophy leading to decreased function and failure is increased wall stress or stiffness. The association of increased interstitial fibrosis with myocardial stiffness in both human and animal studies has suggested a causal relationship [249,250,251,252,253]. Several human [261,262,263,264,265,266,267,268], and experimental animal studies [249,250,251,252,253] have shown that in the early stages of developing hypertrophy from hypertension or valvular stenosis pressure overload, interstitial connective tissue develops in parallel with or following the increasing CM size becoming more pronounced as hypertrophy advances into myocardial failure. However, not all patients or animals with severe hypertrophy progress to failure [264,265,266,267,268,269]. Following the repair of aortic stenosis in both patients and experimental animals, both structural and functional recovery suggests that other factors than the amount of collagen play a role in the complex hypertrophic response [240,241,242,264,270].

### 8.3. Contractile Proteins in Myocardial Failure

Both clinical and experimental animal studies have shown that not only collagen but other cytoplasmic alterations, notably titin, are important factors in myocardial stiffness of hypertrophy [271,272,273]. In the severely hypertrophied and failing myocardium with contractile dysfunction and increased myocardial stiffness, several studies have focused on the relationship of the sarcomere, the contractile structure of the CM, and its relationship to the intercalated disc and the sarcolemmal membrane. Force, generated by the interaction of actin and myosin in the sarcomere, is modulated by the major structural protein of the sarcomere, titin. Titin connects the Z-line with the central M-band, connected end to end from one sarcomere to the next, providing a scaffold for other sarcomere proteins and other cellular functions [274]. Titin plays a major role in controlling myocardial stiffness. Other molecules, including desmin connecting the sarcomere Z-line to the sarcolemma and intercalated discs, microtubules, and others are also altered in failing myocardium. Animal model studies have identified increased myocardial stiffness and alterations of titin, desmin, myosin, actin, microtubules, and others in pressure overloaded failing myocardium [275,276,277,278,279,280,281]. The interaction of these contractile proteins in the failing myocardium has been reviewed by Bernardo, Kruger, and Lyon [282,283,284].

### 8.4. Hemodynamic Effects in Myocardial Failure

The conversion from stable hypertrophy to the decompensated stage with failure involves not only the interstitial components of fibrocytes, collagen, vasculature, mast cells, and the alterations of the sarcomeric proteins described above, but also hemodynamic and subcellular structural alterations in calcium transit, genes, and bioenergetics affecting available energy and contractile performance.

In chronic pressure-induced stable LVH with normal end-diastolic pressures, wall stress is elevated, baseline subendocardial coronary flow is modestly increased and subendocardial coronary reserve is normal [223,224,225,230,232]. Compressive alterations on the subendocardium of the failing myocardium with elevated LV end-diastolic pressure are responsible for the reduced coronary flow reserve in the failing myocardium initiating a series of cellular and subcellular events responsible for the decreased contractile function [223,224,225,230,232]. While hemodynamic changes due to increased wall stress may be responsible for increased cytoskeletal collagen or focal areas of CM necrosis and fibrosis, it is still not clear how such hemodynamic alterations may affect subcellular molecular changes.

### 8.5. Altered Calcium Transit in Myocardial Failure

The T-tubule, sarcoplasmic reticulum, and mitochondrial system control the availability of calcium needed for contraction of the sarcomere. In fetal and neonatal myocardium where cell diameters are small, diffusion of calcium is able to provide adequate calcium for contraction. As CMs increase in diameter during neonatal growth, T-tubules, sarcoplasmic reticulum, and mitochondria are added to allow intracellular calcium for sarcomere contraction and electrical activity [128,130,131]. During normal growth and stable LVH, these calcium regulating structures increase along with increasing CM diameter, and calcium transit is maintained [281,285]. In the failing hypertrophied heart, the T-tubular and sarcoplasmic reticulum is dilated, and more abundant, and calcium transit is compromised [241,285]. Both human [286,287,288] and animal [289,290,291,292] studies have shown altered Ca^2^ ATPase and calcium transit in severe cardiac hypertrophy. Phospholamban and Ca-ATPase are intricately involved in the control of reduced calcium transit in severely hypertrophied and failing myocardium [293]. The various regulators of the sarcolemmal network and calcium transit in myocardial failure have been reviewed by Alpert, Dhalla, and Movesian [294,295,296].

### 8.6. Genes, Isozymes and Protooncogenes in Hypertrophy

In the fetal developing myocardium, there are several growth-regulating factors operating in the reduced oxygen environment of the fetus. As the fetus enters the oxygen saturated phase of growth in the neonatal period, these isozymes, genes, and protooncogenes undergo a shift in structure from the fetal form to the adult form promoting the switch to hypertrophic growth and higher energy demands of the growing heart. The isozyme switch alters their function from promoting hyperplastic fetal growth to supporting stable hypertrophic growth and specifically inhibiting CM division.

During normal adulthood and the early stable period of enhanced overload cardiac hypertrophy, these adult isozymes and protooncogenes continue their function to maintain normal interstitial and cytoplasmic functions including fibroblastic collagen production, calcium transit, sarcomere proteins, ribosomal and Golgi protein production and all other requirements of the myocardium. With advancing severe hypertrophy and development of myocardial failure, many of these adult forms convert to the fetal forms, perhaps in an attempt to assist the overloaded myocardium in a switch from the higher energy requiring the fast form to a more energy-efficient slow form. Whether these conversions of isozymes could assist in allowing the highly differentiated CM to divide is questionable, considering the developed structure of the cell.

Many of these fetal to adult, adult to fetal isozyme, and protooncogenes switches have been discussed in previous sections. Specific examples of clinical and animal studies to evaluate these switches include the myosin heavy chain [297,298,299,300], lactate dehydrogenase [301,302,303], creatine kinase [201,304,305], adenylyl cyclase [306], atrial natriuretic peptide [307], and others. For reviews of the conversion to fetal forms of isoenzymes and protooncogenes in cardiac hypertrophy see [308,309,310,311].

## 9. Mitochondria

### 9.1. Structure and Oxidative Metabolism in Normal and Hypertrophied Myocardium

Cardiac mitochondria have received much attention over the past five decades for their role in not only ATP production regulation but also the regulation of calcium transit and protein synthesis [312]. It has been shown that mitochondrial fission is required for cardiomyocyte hypertrophy mediated by a Ca^2+^-calcineurin signaling pathway [313]. More recent studies using newer methods of mitochondrial isolation developed by Hoppel [314] using a combination of mechanical and enzymatic procedures have allowed more detailed morphologic and functional evaluation in both normal and abnormal myocardium. Two distinct morphologic and functional populations of mitochondria have been identified, mitochondria just beneath the sarcolemma, (SSM) and those adjacent to sarcomeres, (IFM) [56,314,315,316,317]. Hoppel and Riva have identified two distinct forms of the mitochondrial cristae [56]. SSM have predominately lamellar cristae, and IFM predominantly tubular, although both have a mixture of the two forms and it is not clear how cristae structure may affect function. SSM tend to be larger and more variable in shape than IFM. A third group of smaller mitochondria, also labeled IFM, are clustered at nuclear poles, presumably newly formed from nuclear DNA and protein production, ready to move to a subsarcolemmal or fibrillar location. Once formed, new mitochondria increase in size by internal protein production from their mDNA, by fusion with other existing mitochondria, and by fission to form new mitochondria [55,318]. SSM by their location, have the greatest access to oxygen and thus are most active in ATP energy production. IFM lying adjacent to the contracting sarcomere are more efficient in transferring energy to myofilaments and their ends lying adjacent to the sarcomeric reticulum, are more active in calcium transit. Presumably, ATP formed by SSM is somehow transported to IFM for the supply of energy to the sarcomere, whether by some physical connection or by cytoplasmic diffusion is not clear. The extensive microtubular system throughout the cytoplasm could be the means of transporting ATP from SSM to IFM [318].

Older studies had calculated mitochondrial and myosin proteins’ half-life at 5–6 days [139,319], but newer studies indicate that the many mitochondrial proteins have different half-lives, some as short as 3–5 days, others as long as 40–50 days, and mDNA about 250 days [315]. Mean half-lives have been calculated at 20 days [316] or 30 days [315] by different investigators. SSM and IFM have different turnover times among the many different mitochondrial proteins with the mean half-lives about equal [315,316].

In a rat model of aortic constriction stable hypertrophy, protein turnover was more in IFM than SFM, increased in some, and decreased in others, resulting in no change in mean half-life due to hypertrophy [316]. The results of many studies regarding the respiratory activity of isolated mitochondria have varied from depressed function, normal function, and increased function in stable hypertrophy, and either normal or depressed function in hypertrophy with failure. The varying results due to different animal models and isolation techniques have been reviewed by several authors [320,321,322]. Rosca and Hoppel [321] have reviewed the effects of various forms of pathology on mitochondria. Ischemia affects primarily the SSM causing vacuolation, while pressure overload hypertrophy more severely affects IFM.

Decreased levels of high energy phosphate flux rate have been reported in patients with LVH due to aortic stenosis [323], and in numerous animal models of pressure overload or myocardial remodeling after severe ischemic injury [289,324,325,326,327]. In early mild LVH, creatine phosphate/ATP levels are unaltered [289] but become depressed as hypertrophy progresses to advanced levels and myocardial failure. With advanced LVH and failure, ATP and creatine phosphate are reduced and ADP is increased to maintain the driving force of the mitochondria ATP production, which is accompanied by an increased glycolytic metabolism [328,329,330,331,332]. The bioenergetic abnormalities occurring in the various forms of patient and animal cardiac hypertrophy have been discussed in detail by Zhang and coauthors [324,325,326].

### 9.2. Mitochondrial Morphology in Hypertrophy and Failure

Quantitative ultrastructural studies of mitochondrial structure, number, and size have supported the original observations and progression of Meerson [187]. The three stages of developing cardiac hypertrophy and failure indicated by Meerson include an initial hyperactive phase with overproduction of mitochondria, a second stage with normalization of the mitochondria paralleling the increase in myocyte size, and a third stage of decompensation with decreased mitochondrial content and smaller sized mitochondria. Subsequent morphometric analysis of animal models with progressive constriction of the ascending aorta or pulmonary artery, perinephritic hypertension, strenuous exercise, or SHR during stable hypertrophy reported either no change or a mild decrease in the mitochondrial cell volume and size [251,281,333,334]. Mitochondrial volume percent is decreased but there are increased numbers of small mitochondria. With developing myocardial failure, mitochondrial volume percent is further decreased with increasing numbers of small, apparently newly formed, mitochondria. Qualitative evaluation of severely hypertrophied and failing human and animal models has reported the presence of large numbers of small mitochondria, often with disruptions of myofibrillar structure [172,175,195,205,209,241] (Figure 8). Functional evaluation of different sized mitochondria has been difficult due to variable techniques for mitochondrial isolation. Are the very small mitochondria presumably produced by fission from stressed mitochondria in failing myocardium capable of the normal functions of protein synthesis and oxidative phosphorylation?

## 10. Sarcomerogenesis in Growth and Hypertrophy

### 10.1. Normal Growth

As CMs leave the fetal stage of hyperplastic growth and all further growth is by an increase in size, new sarcomeres are added to increase both the length and width of the CM. As discussed above in Section 2, during fetal hyperplastic growth incompletely formed sarcomeres are fully disassembled during mitosis and then reassembled in the daughter cells [4,38,45]. New myofibrillar proteins are produced by the ribosomes and Golgi for the daughter cells. After the completion of hyperplastic growth as cells enter hypertrophic growth, sarcomeres become fully formed and are no longer able to be disassembled. Newly formed sarcomeres are stable and maintain their same length throughout the rest of life. For the cells to increase in length during normal growth and in physiologic or pathologic hypertrophy additional sarcomeres need to be added at the ends or within the myofibril. To add width to the CM, new myofilaments must be added to increase the width of existing sarcomeres. Further, the sarcomere is a dynamic structure as the myofilaments have a half-life of from 3–10 days for different proteins [139,335]. Newly formed myofibrils must be inserted into the actively contracting sarcomere and the worn-out proteins removed. The role of microtubules, mitochondria, Golgi, endoplasmic reticulum, growth factors, and other cellular elements in this continuing replacement and addition process has received much attention in recent years [318,336,337,338].

### 10.2. Microtubules

The role of microtubules in the maintenance of CM structure and function has renewed interest since the earlier ultrastructural recognition in CM by Goldstein and others [339,340,341]. These earlier studies identified short microtubules and reported increased microtubules during the early rapid CM growth with the initiation of hypertrophic growth, then stable numbers into adulthood [339]. Microtubules were reported to be increased during stable hypertrophy in a pressure overload guinea pig model and further increased during myocardial failure [276]. Electron microscopy studies suggested a microtubule network, but the true nature required immunostaining and confocal microscopy to elucidate the extensive microtubule and intermediate filament network and discover functional roles in controlling sarcomere force, transmitting mechanical signals, myofilament renewal and removal, mitochondrial maintenance and other cell maintenance functions [318]. The three interlocking domains of microtubules are located around (a) the nucleus, (b) around the myofibers and sarcomeres, and (c) interfibrillar organizing T-tubules and sarcoplasmic reticulum. For a review of the extensive microtubule and intermediate filament structure and function in normal and hypertrophied failing hearts see Caporizzo [318]. The microtubular system surrounding the myofibrils is increased during stable pressure overload hypertrophy [276,337], and much further increased in the failing myocardium [276,338] contributing to the increased viscous load on contracting myofilaments.

### 10.3. Intercalated Disc and Z-Bands

Ultrastructural and autoradiographic studies of fetal and neonatal CM demonstrated that lateral accumulations of dense filament attachment sites (costameres, fascia adherens junctions, Z-band material) are the site for new protein accumulation into Z-bands and production of sarcomeres producing myofibrils under the CM sarcolemma [18,38,47,53,54]. As the CM transit to hypertrophic growth, these lateral dense sites migrate to the ends of the CM forming mature intercalated discs (ICD) [18,342] (Figure 3). As the cells increase in length, new sarcomeres are added from the ICD at the rate of about one sarcomere per day [342]. With increasing growth, the lateral subsarcolemmal myofibrils migrate inward as new myofibrils are added at the periphery and from the ICD. Scanning electron microscopy of isolated myocytes from rapidly growing and hypertrophied CM illustrates the increased complexity as ICDs are added to the cell. In neonatal CM the cross-section of new myofibrils is roughly circular but with continued growth, the myofibrils merge into a confluent mass filling the entire cytoplasm (Figure 9). Throughout normal growth, adult life, and stress-stimulated hypertrophy, the intricate hexagonal array of myofilaments is maintained as old myofilaments are replaced with new ones.

How are new myofilaments produced in overload cardiac hypertrophy? The ICD is the source for adding sarcomeres at the ends of myofibrils. Actin thin filaments are attached to filament attachment sites, fascia adherens, and incompletely formed new sarcomeres are adjacent. The ICD in stable adult myocardium has a narrow, wavy course less than 0.3 µm wide (see Figure 5). During the rapid growth phase of young animals [342] and in the severely stressed hypertrophied and failing myocardium [333] the normally narrow ICD becomes up to 2.0+ µm wide with folds and convolutions. Within the ICD folds, newly formed actin and myosin filaments appear and mature into fully formed sarcomeres. Convoluted folding of the ICD in addition to that reported in rapidly growing animals [342] is also a feature of severely hypertrophied and dilated myocardium in dogs with progressive pulmonary artery stenosis [333] , one-year-old SHR [238], and patients with IHSS [172] (Figure 10).

In addition to new sarcomeres being added to the ends of myofibrils in severely hypertrophied and failing myocardium, alterations of the Z-band occur producing newly formed sarcomeres within the myofibril [167,172,239,333]. The Z-band consists of a complex lattice of proteins that anchor both titin and actin filaments of the sarcomere and provides not only a mechanical role in transmitting the developed force [343] but also is intricately involved in sarcomerogenesis [333,344]. In fetal, neonatal, and adult myocytes, Z-bands are attached to the dense Z-band material along the sarcolemma and the ICD. Thin actin filaments appear first, soon followed by myosin, myomesin, titin, and other filaments to form a complete sarcomere.

Expanded forms of the Z-band ranging from mild thickening to variations up to 1.5 µm width have been described in both animals with severe pressure overload hypertrophy [333,343,344,345], old SHR [167,239], and in patients with dilated cardiomyopathy [205,344] and IHSS [172]. The expanded Z-lines appear to be inserting a new sarcomere within the myofibril forcing the affected fibril to be out of line with adjacent sarcomeres thereby increasing the length of the myofibril (Figure 11).

In severely volume overloaded experimental animal or human hearts, CMs become elongated with little increase or even decrease in width, depending on cell volume change, resulting in slippage of myocytes discussed above [202]. The resulting misalignment of sarcomeres and disruption of Z to Z-band connections (see Figure 6) would appear to be at least partially responsible for the reduced contractile function of these failing hearts.

Pressure overloaded hearts increase cell cross-sectional area with no increase in cell length, requiring myofilaments to be added to existing sarcomeres. Cross-sections of pressure overloaded CM reveal an expanded confluent myofiber mass with a normal hexagonal arrangement of myofilaments. This increased accumulation of myofilaments is accomplished while maintaining the removal of the rather short half-life filaments. The balance of sarcomere filament addition and removal is only partially understood, involving the endoplasmic reticulum, mitochondria, and the extensive microtubular system [318,344,346].

## 11. How Do Stressed Cardiomyocytes Adapt, Degenerate, and Die?

The most serious injury to the myocardium is acute myocardial ischemia due to obstruction of a major coronary artery, with death to the affected cells within minutes to hours if reflow is not reestablished [347]. Ischemic cells die by the classical oncotic coagulation necrosis and fibrocystic repair. Lesser amounts of reduced oxygen availability as with small vessel occlusions [348] or short or long-term coronary flow/stress-induced oxygen demand will cause small focal areas of ischemic necrosis ending with focal areas of fibrosis [190,241,348]. The process of necrotic cell death due to ischemia is characterized by nuclear and cytoplasmic changes, rupture of the cell membrane with the release of contents to the cytoplasm inciting an inflammatory response ending in fibrosis. Examples of this type of cell death include aortic or subaortic stenosis with a coronary flow imbalance between the low-pressure coronary artery and the high-pressure LV [240,241,242], sudden acute experimental occlusions of the aorta or pulmonary artery [243,244], advanced LV hypertrophy with additional stress such as imposed exercise [232], and the imbalance of coronary flow with stress requirements in late-stage LVH and failure [215,230].

Apoptosis is a process of cell death without membrane rupture, accumulation of apoptotic bodies within the dying cell, and eventual removal by phagocytosis without an inflammatory response. The role of apoptosis in myocardial remodeling during normal heart growth, developing hypertrophy, dilated cardiomyopathy and ischemic heart disease has been studied in numerous publications for human [349,350], and experimental animal [348,351,352] studies among others. Techniques to identify apoptosis include the identification of DNA strand breaks with the TUNEL procedure which, when combined with enzyme procedures for caspase and Bcl-2, improves the accuracy [353]. The use of TUNEL alone in earlier studies caused much confusion, with some studies reporting very high levels of apoptosis. The time course of the apoptosis process in the myocardium is not known but has been estimated at a few hours to days. If the time for apoptosis is short, and the high rates of apoptosis were correct, it would not take long for the heart to melt away. The problems with the use of TUNEL alone were pointed out by Ohno [354] and discussed in a review of the issue by Buja [355]. Ischemic CMs, as well as non-myocytes undergoing necrosis or apoptosis, will react positively to TUNEL staining. The problem with determining the number of CMs staining positively with any of the techniques is not trivial. First, the usual positive rate is very low, 1-3 percent, requiring a very large field to be analyzed. Secondly, interstitial endothelial, fibrocytes and other cells lie in close apposition to the CM, making separate identification difficult. Very few studies have used morphologic techniques to supplement TUNEL or other histochemical methods to verify true apoptosis. Morphologically identified apoptotic cells are generally located within dense fibrous tissue adjacent to ischemic induced fibrosis [348]. Additional reviews have discussed the application and interpretation of apoptosis in myocardial hypertrophy and injury [311,353].

Necrosis and apoptosis result in total cell loss and replacement by focal fibrosis or removal by phagocytosis. Severely stressed cells during the progression from stable hypertrophy to myocardial failure undergo degenerative changes affecting the nucleus, mitochondria, sarcoplasmic reticulum, sarcomere, and other organelles. Ultrastructural nuclear changes include clumping of chromatin, undulations of the nuclear membrane, and dense chromatin clumping along the nuclear membrane. The sarcoplasmic reticulum and T-tubules become dilated. Mitochondria may become vacuolated with loss of cristae and many small mitochondria predominate. In severe late stages, the sarcomeres are disrupted into free filaments [172,205]. Many of these degenerative changes are reversed if the hemodynamics can be returned to normal as with aortic valve replacement [241]. A characteristic feature of hypertrophied CM in addition to the increase in cross-sectional area and perhaps length is the marked distortion from the nearly circular shape of growing and adult normal CM. Hypertrophied CM develop multiple connections with adjacent myocytes and enlarge to envelop capillaries, even forming tunnel capillaries [123,239]. Scanning electron microscopy of isolated myocytes from severely hypertrophied myocardium also reveals a marked disparity in CM size, some very large, others very small [123] (Figure 12). Whether the small cross-sections are the result of CM branches or due to degenerative loss of contractile material is not clear. It is conceivable that with continued hemodynamic and physical stress on the CM, the continual balance of myofilament degradation and renewal under the control of the microfilament system described above could result in a gradual decrease in sarcomeric filaments resulting in smaller, yet still viable CM. Though difficult to evaluate, the marked variability in cell size of severely hypertrophied CM would likely contribute to the decreased contractile function.

## 12. Conclusions

The dynamic development of the CM from its earliest contractile stage in the fetus, through birth and neonatal growth, adolescent to adult growth, and the response to physiologic and pathologic stress involves the interaction of the CM with interstitial components as well as dramatic intracellular changes. Structural and functional development involves interaction with many genetic, biochemical, growth factors, metabolic and other factors. The changing isoform nature of many of these factors controls the dramatic alteration from cells dividing to increase the mass of the heart to cells converting to only hypertrophic growth and unable to divide, and finally to the development of pathologic alterations resulting in myocardial failure. The interaction of interstitial connective tissue, coronary vasculature, and other interstitial cells play a role in the normal development as well as the pathological changes resulting in decreased contractile function and myocardial failure. The constantly evolving technology over the past century has provided an increased understanding of structural, biochemical, and functional features of the heart which have led to a better knowledge of both CM and interstitial cardiac growth, yet unresolved questions remain.

## Figures and Tables

**Figure 1 biology-11-00880-f001:**
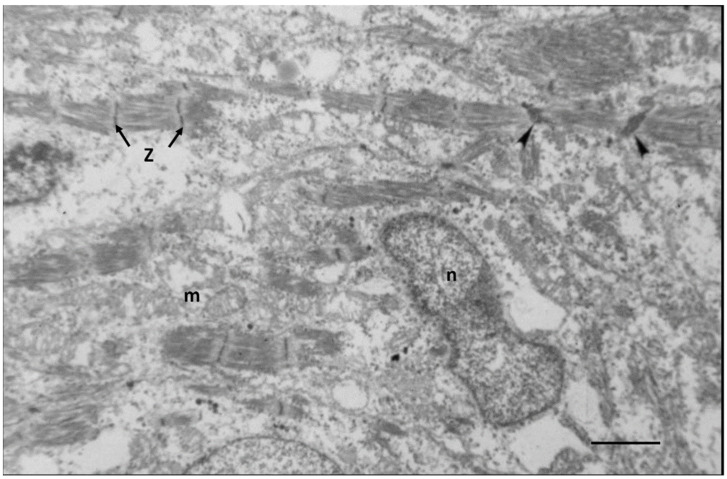
Electron micrograph of an E12 rat embryo. The nucleus (n) is immature with dispersed chromatin, mitochondria (m) are rare, endoplasmic reticulum is abundant. Myofiber fragments have immature sarcomeres bounded by Z-bands (z). Wide Z-bands (arrowheads) suggest neoformation. Bar = 2 µm.

**Figure 2 biology-11-00880-f002:**
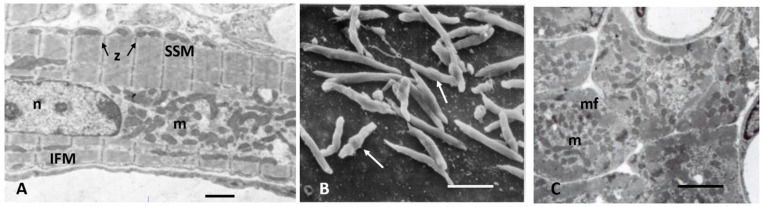
Neonatal rat. (**A**) Electron micrograph of P8 rat. Lateral sub-sarcolemmal myofibrils aligned between Z-bands (z), mitochondria (m) adjacent to the nucleus, intermyofibrillar (IFM), and subsarcolemmal (SSM). Bar = 2 µm. (**B**) Scanning electron micrograph of P3 rat isolated cardiomyocytes; note spindle shape and central nuclear bulge (arrows); Bar = 20 µm. (**C**) Electron micrograph of cross-section cardiomyocytes of P6 dog. Note peripheral myofibrils (mf), central mitochondria (m). Bar = 2 µm.

**Figure 3 biology-11-00880-f003:**
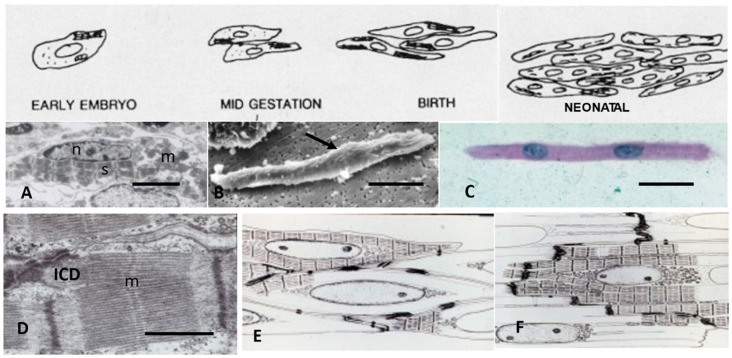
(**A**) Fetal cardiomyocyte with immature sarcomeres (s), mitochondria (m), and nucleus (n). (**B**) Isolated newborn cardiomyocyte with a single nucleus (arrow). (**C**) Neonatal isolated cardiomyocyte with two nuclei. Bar in A = 5 µm, Bar in B and C = 10 µm. (**D**) Electron micrograph of P10 rat. Myofilaments attached to intercalated disc adherens junction (ICD). Sarcomere complete with M-band (m). Bar = 1 µm. (**E**) Schematic drawing of P1 spindle-shaped neonatal cardiomyocytes. (**F**) Schematic drawing of P10 neonatal cardiomyocytes. Lateral adherens junctions (dark) move to the cell end to form the ICD.

**Figure 4 biology-11-00880-f004:**
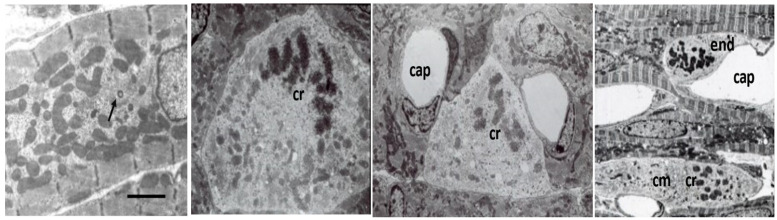
Electron micrographs of P3-P8 rat cardiomyocytes (cm). Left panel, subsarcolemmal fibrils, variable-sized mitochondria, and centrosome (arrow). Bar = 2 µm. Panels with mitoses with clumped chromatin (cr), dispersed myofibrils small mitochondria, and pale cytoplasm. Capillaries (cap) are dilated due to perfusion fixation. Endothelial cell mitosis (end).

**Figure 5 biology-11-00880-f005:**
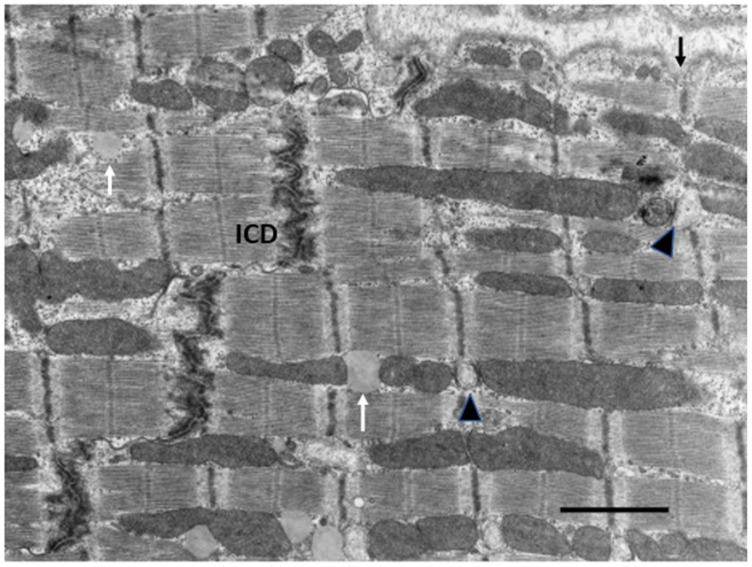
Electron micrograph of a normal adult rat. Note sarcomeres attached to intercalated disc (ICD) and sarcolemma at Z-band (arrow). Sarcoplasmic reticulum at Z-bands adjacent to mitochondria (arrowheads). Lipid (white arrow) Bar = 2 µm.

**Figure 6 biology-11-00880-f006:**
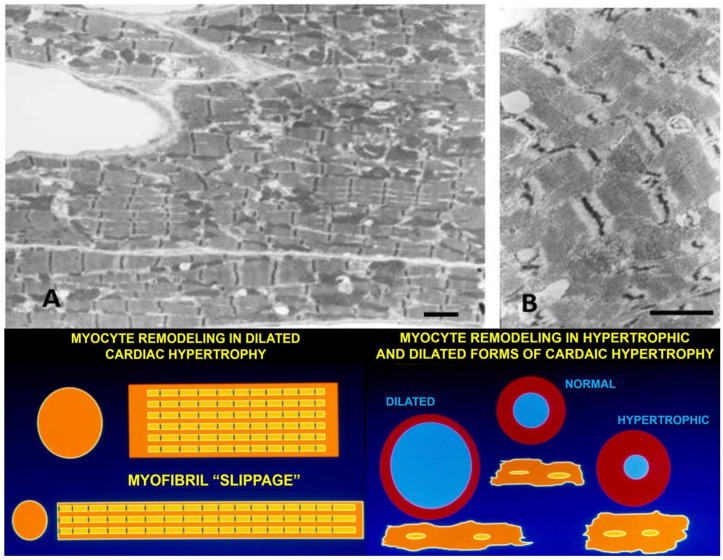
(**A**,**B**) Electron micrographs of myocardium from a dog with pacing-induced eccentric hypertrophy and increased end-diastolic pressure. Due to induced stretch, sarcomeres have lost their normal alignment within myofibrils. Compare with Figure 5. Bar = 2 µm in each. Lower panels: schematic illustrations of myofiber rearrangement and cell size change in dilated and hypertrophic cardiomyopathy.

**Figure 7 biology-11-00880-f007:**
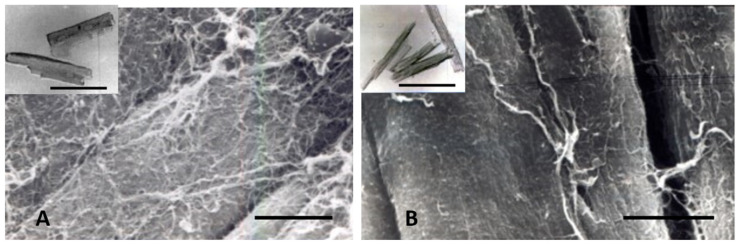
Scanning electron micrographs of (**A**) normal adult dog, and (**B**) dog with chronic experimental mitral insufficiency. Loss of interstitial connective tissue in a dog with LVH and myocardial failure. Bar = 10 µm in both. Insets: isolated myocytes from normal and LV failure dog. Myocytes are longer and thinner in mitral regurgitation with failure. Bar = 100 µm in both.

**Figure 8 biology-11-00880-f008:**
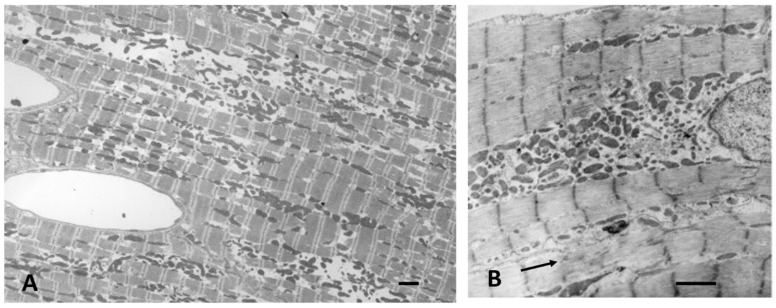
Electron micrographs of dilated LV from dogs with chronic mitral regurgitation and congestive heart failure. (**A**) There are many small mitochondria and focal distortions of myofibrils and sarcomeres in (**B**) (arrow). Bar = 2 µm in each.

**Figure 9 biology-11-00880-f009:**
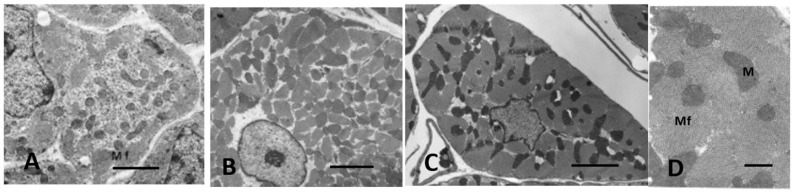
Electron micrographs of cross-sectioned LV myocytes from dogs. (**A**) neonatal, (**B**) normal adult, and (**C**) pressure overload LVH. (**D**) higher power myofilaments LVH myocardium. Myofibrils coalesce into a confluent mass with increasing hypertrophy. Mf = myofibril. M = mitochondria. Bars = 4 µm in (**A**–**C**), 2 µm in (**D**).

**Figure 10 biology-11-00880-f010:**
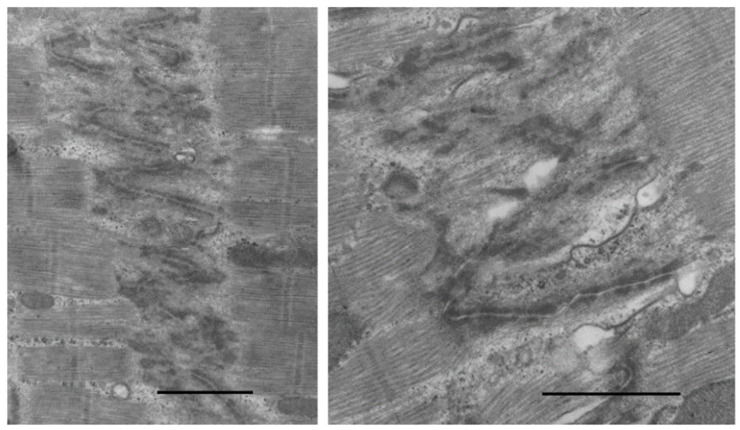
Electron micrographs of expanded ICD in dogs with progressive pressure overload LVH and failure from aortic banding as puppies. Increased folding of ICD has actin and myosin within folds forming new sarcomeres. Bar = 2 µm in each.

**Figure 11 biology-11-00880-f011:**
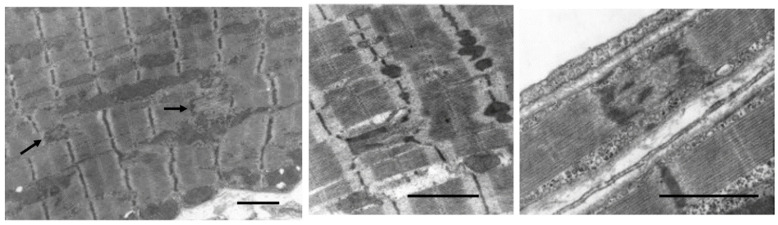
Electron micrographs of RV from a dog with progressive stenosis of the pulmonary artery, RVH, and congestive failure. Z-band expansions (arrow) forming a scaffold for insertion of a new sarcomere within the myofibril and malalignment of myofibril Z-bands. Bar = 2 µm in each.

**Figure 12 biology-11-00880-f012:**
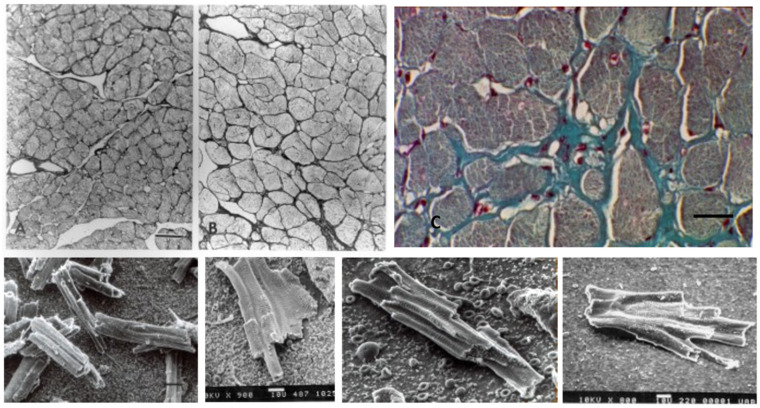
(**A**) Sham-operated and (**B**) 8-week aortic banded rats at same magnification. Note increased size and variable cross-sectional area in the banded rat. Methacrylate embedded, 1 µm section, silver stain. Bar = 50 µm for both A and B. (**C**) Patient with old ischemic infarct. Note marked variability in hypertrophied LV myocytes. Masson trichrome stain. Bar = 20 µm. (**Bottom row**) Scanning electron micrographs of isolated myocytes from hypertrophied LV of aortic banded rats. Note extreme variability in size and shape, grooves for capillaries.

## Data Availability

Not applicable.

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
