# Peer review of "Cardiomyocyte Proliferation from Fetal- to Adult- and from Normal- to Hypertrophy and Failing Hearts"

_biology, 2022, doi:10.3390/biology11060880_

Round 1

Reviewer 1 Report

This is an extensive review manuscript that discusses many aspects related to changes that happen in the heart from fetal to neonatal development and into adulthood.

Main concerns:

  • Most topics are discussed in broad terms without going into too much detail, resulting in a relatively shallow review of (too?) many aspects of how cardiomyocytes develop and adapt to stress.
  • The first author has recently published a similar review article with largely overlapping topic.
  • Most of the references are dated with the majority originating from the previous millennium, which is not necessarily a bad thing, but in lieu of more recent insights into the various areas that are covered in the review, it actually does matter. The consequence is a mostly dated review article. This is further corroborated by an almost exclusive use of electron microscopy to illustrate certain aspects that are pointed out in the review.
  • For the EM images, it is likely immediately obvious what one should be looking at and how this is different from normal if one evaluates EM images on a daily basis. However, for a reader without a background in EM imaging, some arrows or other indicators to point out what the authors wants the reader to focus on would be very helpful.
  • The review is overly long and repetitive at points.

Reviewer 2 Report

The topic of the manuscript is very well discussed. The Authors provide a detailed review that nuclear mitosis and cell division are involved in hyperplastic and hypertrophic growth during fetal to neonatal period. In the controlling of genetic, metabolic biochemical and other factors, pressure overload, volume overload, hemodynamic and biochemical stress lead to structural and functional alterations of CM resulting in hypertrophic, myocardial failure, following increase the mass of the heart.

Major revision:

-Cardiac mitochondrial Reactive oxygen species (ROS) is strongly associated with cardiac hypertrophy. What is role of ROS in hypertrophy of CM during fetal to neonatal period?

Minor revision:

-Which Biomarkers are used for fetal to neonatal hypertrophy?

- Quality of Figure 3. D) should be improved, in some cases there are difficult to read.

- Line 353-354, “µm3” should be “µm3

Reviewer 3 Report

Samford P. Bishop and Lei Ye wrote a useful review giving the reader an overview of growth of the cardiomyocytes from fetal to hypertrophy and failure. The review is very comprehensive, showing all aspects of the cardiomyocytes growth and hypertrophy, taking into account species differences. All the important resources were referenced to databases. The better knowledge of normal as well as pathological processes involved in the cardiomyocytes growth might be helpful in finding new therapeutic approaches specifically tailored to pathological hypertrophy in human heart failure.

Author Response

Thank you for your positive review of our manuscript.

Round 2

Reviewer 1 Report

No more comments.